# Role of lncRNA XIST/miR-146a Axis in Matrix Degradation and Apoptosis of Osteoarthritic Chondrocytes Through Regulation of MMP-13 and BCL2

**DOI:** 10.3390/biology14030221

**Published:** 2025-02-20

**Authors:** Sara Cheleschi, Nicola Mondanelli, Iole Seccafico, Roberta Corsaro, Elena Moretti, Giulia Collodel, Antonella Fioravanti

**Affiliations:** 1Rheumatology Unit, Department of Medicine, Surgery and Neuroscience, Azienda Ospedaliera Universitaria Senese, Policlinico Le Scotte, 53100 Siena, Italy; saracheleschi@hotmail.com (S.C.); ioles388@gmail.com (I.S.); 2Section of Orthopedics and Traumatology, Department of Medicine, Surgery and Neurosciences, University of Siena, Policlinico Le Scotte, 53100 Siena, Italy; nicola@nicolamondanelli.it; 3Department of Molecular and Developmental Medicine, University of Siena, 53100 Siena, Italy; r.corsaro@student.unisi.it (R.C.); elena.moretti@unisi.it (E.M.); 4Independent Researcher, 53100 Siena, Italy; fioravanti7@virgilio.it

**Keywords:** long non-coding RNA, microRNA, osteoarthritis, chondrocytes, apoptosis, metalloproteinases

## Abstract

Osteoarthritis is the most frequent chronic and degenerative musculoskeletal disorder affecting the adult and elderly populations worldwide, leading to pain and disability. It has been observed that the prevalence of the disease has increased exponentially in recent years, representing an important health burden with relevant implications for the public health system. Osteoarthritis is a complex condition due to different factors. It leads to the progressive degradation of cartilage associated with an alteration in the other structures that constitute joint tissue. A fundamental goal of many scientists working in this field is to better comprehend the mechanisms underlying the pathogenesis of osteoarthritis to improve its diagnosis and therapy. Our results provide evidence about the role of long non-coding RNAs and microRNAs, important epigenetic factors, in the development and progression of osteoarthritis; in particular, we demonstrate that these molecules can act together to regulate cell viability, cell death, and cartilage degradation, the major hallmarks of osteoarthritis. These findings can help to improve the knowledge about the molecular mechanisms implicated in osteoarthritis damage, identify therapeutic targets, and develop specific therapies useful in counteracting the disease.

## 1. Introduction

Osteoarthritis (OA) is the most frequent chronic and degenerative musculoskeletal disorder affecting the elderly population worldwide, leading to pain and disability [1]. As estimated by the Global Burden of Disease study 2019, the prevalence of OA has exponentially increased in recent years, representing an important health burden with relevant implications for clinical and public health systems [2]. OA is a complex condition with a multifactorial etiology that leads to progressive cartilage degradation accompanied by biochemical and metabolic changes in joint tissues [3].

During OA onset and development, articular cartilage is subjected to structural remodeling managed by different factors such as mechanical stresses, genetic predisposition, and low-grade inflammation. Chondrocytes turn into a hypertrophic-like phenotype, resulting in a modified production of inflammation-related genes and catabolic genes and in alterations in different pathways linked to the extracellular matrix (ECM) structure; the consequence of the aberrant production of matrix-degrading enzymes is the continued modification of the structure of articular cartilage and of its normal functioning. These alterations cause the progressive destruction of articular cartilage, inflammation of the synovial membrane, thickening of subchondral bone, and formation of osteophytes [4,5].

However, the exact mechanisms underlying the pathogenesis of the disease have not been fully clarified.

A growing number of studies have shown that epigenetic factors, including non-coding RNAs, are closely related to the pathological processes occurring in OA [6,7,8,9,10]. Among them, long non-coding RNAs (lncRNAs) have been identified as a group of RNA molecules of over 200 nucleotides in length that interact with DNA and RNA sequences and protein molecules controlling the expression level of target genes via epigenomic, transcriptional, or post-transcriptional approaches. LncRNAs actively take part in the regulation of many biological processes and pathological events, acting as regulatory signals, guiding factors, decoys, and scaffolds [11,12]. These transcripts also operate as competitive RNAs, which serve as molecular sponges for microRNAs (miRNAs) to release miRNAs target genes, exerting a post-transcriptional regulation of their expression in different molecular mechanisms [11,13]. Recently, the implication of lncRNAs as new regulators of the progression of OA damage has been proven. Indeed, the results derived from a microarray analysis found that a large number of lncRNAs were differently expressed in OA cartilage compared with normal specimens [14,15,16,17], specifically regulating cell proliferation, differentiation, and apoptosis, as well as inflammation and ECM production [12,18,19,20,21,22,23].

The lncRNA X-inactive-specific transcript (XIST) is one of the most commonly studied lncRNAs, and it is widely recognized as an important regulator of cell growth and a common oncogene that participates in the onset of tumors and other human diseases [24]. Recent evidence has affirmed that the lncRNA XIST is also strongly associated with the development of OA [12,24]. It has been reported that it is involved in regulating the proliferation and apoptosis of OA chondrocytes through MAPK signaling via miR-211 [25]; in addition, the up-regulation of XIST was observed to promote extracellular matrix degradation through the activation of metalloproteinase (MMP)-13 in OA samples and articular chondrocytes extracted from OA tissues and interleukin (IL)-1β-exposed cells; on the other hand, its silencing was found to be able to inhibit the destruction of the ECM via sponging miR-1277-5p [26]. Another in vitro study demonstrated the inhibition of the ECM degradation by XIST knockdown, heightening the protein levels of collagen and aggrecan in IL-1β-treated chondrocytes [27].

However, the recent data from the literature indicate that the exact mechanisms by which XIST exerts its functions in different OA processes remain still largely unknown and deserve further investigation. Therefore, the present research aimed to analyze the role of the lncRNA XIST in the pathogenesis of OA in an in vitro study on human OA chondrocyte cultures in the presence or not of the negative stimulus of IL-1β. In particular, we evaluated the effects of XIST on cell viability, apoptosis, and ECM degradation, known as the major processes involved in OA damage; furthermore, we studied the possible interaction of XIST with miR-146a, which is considered one of the main miRNAs implicated in the onset and progression of the disease due to its activity as regulator of cartilage turnover and apoptosis signaling. Finally, we studied the potential molecular mechanism implicated in XIST functions through an analysis of the nuclear factor (NF)-κB signaling pathway.

## 2. Materials and Methods

### 2.1. Sample Preparation and Cell Cultures

Human OA articular cartilage was obtained from femoral heads of five patients (two men and three women, from 64 to 77 years old, with body mass index of 20 to 26 kg/m^2^) with coxarthrosis defined by the ACR criteria [28] and subjected to surgery. The specimens were supplied from the Orthopaedic Surgery of the University of Siena (Italy). OA grades ranged from moderate to severe according to Mankin score [29]. Permission to use human samples was provided by the Ethic Committee of Azienda Ospedaliera Universitaria Senese (decision No. 13931/18) after obtaining informed consent from each donor.

The cartilage was removed from the femoral head, cut into small pieces, and processed using enzymatic digestion through tripsin (Euroclone, Milan, Italy) for 30 min and type IV collagenase (Euroclone, Milan, Italy) for 16 h [30]. The obtained cells were grown in Petri plates of 100 mm with Dulbecco’s modified eagle medium (DMEM) (Euroclone, Milan, Italy) containing phenol red and 4 mM L-glutamine (Euroclone, Milan, Italy) and supplemented with 10% fetal bovine serum (FBS) (Euroclone, Milan, Italy) and 200 U/mL penicillin and 200 µg/mL streptomycin (P/S) (Sigma-Aldrich, Milan, Italy). The medium was replaced once a week, and the growing cells were checked with an inverted microscope (Olympus IMT-2, Tokyo, Japan). After evaluating the viability with Tripan Blue test (Sigma-Aldrich, Milan, Italy), primary cells were employed for the experiments.

Human immortalized T/C-28a2 chondrocyte cell line was provided by the laboratory of Goldring Group and derived from a human juvenile costal cartilage. After the enzymatic digestion, the isolated cells were subjected to immortalization carried out by a transfection procedure by a retroviral vector pZipNeoSV (X) and the SV40 large T antigen. T/C-28a2 cells were used as control culture [31].

### 2.2. Cell Transfection

Chondrocytes were plated in 6-well dishes at a starting density of 1 × 10^5^ cells/well in DMEM supplemented with 10% FBS, which was then substituted with 0.5% FBS for a period of 6 h until the transfection procedure. After that, cells were transfected with small interfering RNA (siRNA) against XIST (Qiagen, Hilden, Germany) (50 nM) or with a specific miR-146a inhibitor (Qiagen, Hilden, Germany) (50 nM) in serum-free medium for 24 h. Cells used as controls were incubated with the negative control siRNA (NC) (Qiagen, Hilden, Germany) (5 nM) [32]. The treatment was performed in presence or not of the negative stimulus of IL-1β at the concentration of 10 ng/mL.

After the transfection, the medium was discharged, and the cells were immediately collected for the following analysis.

### 2.3. Cell Viability

Cells were plated in 12 well-plates (8 × 10^4^ cells/well) for 24 h in DMEM with 10% FBS before substitution with 0.5% FBS used for the transfection procedure. After that, the viability of the cells was evaluated using 3-[4,4-dimethylthiazol-2-yl]-2,5-diphenyl-tetrazoliumbromide (MTT) test (Sigma-Aldrich, Milan, Italy), as already reported [33].

The percentage of survival was determined as (absorbance of considered sample)/(absorbance of control sample) × 100 using a microplate reader spectrophotometer at 570 nm (BioTek Instruments, Inc., Winooski, VT, USA). Data were expressed as OD units per 10^4^ adherent cells.

### 2.4. Flow Cytometry Assessment

For this analysis, chondrocytes were plated in 12-well plates (8 × 10^4^ cells/well) for 24 h in DMEM with 10% FBS, which was then substituted with 0.5% FBS used for the transfection.

The evaluation of apoptotic cells was performed using a commercial kit equipped with Annexin V and propidium iodide (PI) (ThermoFisher Scientific, Milan, Italy) probes. Afterward, the cells were washed, collected using trypsin, posed into cytometry tubes, and centrifuged at 1500 rpm for 10 min. A total of 100 μL of 1 × Annexin-binding buffer, 5 μL of Alexa Fluor 488 annexin-V conjugated to fluorescein (green fluorescence), and 1 μL of 100 μg/mL PI (red fluorescence) working solution were added to the pellet of the cells and incubated for 15 min in the dark. Then, 600 μL of 1 × Annexin-binding buffer was added before the analysis. A total of 10,000 events for each experimental condition were measured using flow cytometer (Cy Flow Cube 6, Sysmex Partec, Milan, Italy). The results were elaborated using Cell Quest software (Version 4.0, Becton Dickinson, San Jose, CA, USA).

Apoptosis was detected considering the staining cells simultaneously with Alexa Fluor 488 annexin-V and PI, representing the percentage of the cells positive to each dye (total apoptosis). Data were reported as the mean of three independent experiments.

### 2.5. RNA Isolation and Quantitative Real-Time PCR

Cells were grown in 6-well dishes at a starting density of 1 × 10^5^ cells/well in DMEM supplemented with 10% FBS before the treatment procedure. Then, the cells were collected, and total RNA was extracted using TriPure Isolation Reagent (Euroclone, Milan, Italy) according to the manufacturer’s instructions. The quality of the RNA was assessed by measuring the OD at 260 nm and the 260/280 and 260/230 ratios using a Nanodrop-1000 (Celbio, Milan, Italy).

Reverse transcription for the target genes was carried out using the QuantiTect Reverse Transcription Kit (Qiagen, Hilden, Germany), reverse transcription for lncRNAs was carried out using RT^2^ First Strand Kit (Qiagen, Hilden, Germany), and reverse transcription for miRNAs was carried out using the cDNA miScript PCR Reverse Transcription Kit (Qiagen, Hilden, Germany).

Target genes, lncRNAs, and miRNAs were analyzed via the real-time PCR using QuantiFast SYBR Green PCR (Qiagen, Hilden, Germany), RT^2^ SYBR Green (Qiagen, Hilden, Germany), and miScript SYBR Green (Qiagen, Hilden, Germany) kits, respectively. Appendix A reports the list of primers used for PCR.

The reactions were achieved using a LightCycler 1.0 (Roche Molecular Biochemicals, Mannheim, Germany) with LightCycler Software Version 3.5 [34].

For data analysis, the Ct values and efficiency of the primer set were calculated and converted into relative quantities [35,36]. For data normalization, actin beta (ACTB), U6, and Small Nucleolar RNA, C/D Box 25 (SNORD-25), were used as housekeeping genes [37,38].

### 2.6. Immunofluorescence Analysis

For this determination, chondrocytes were plated and grown in sterile coverslips inserted in multi-wells at a starting low density of 4 × 10^4^ cells/chamber to prevent a complete cell confluence or possible overlapping. Cells were incubated with the specific inhibitors for XIST and miR-146a for 24 h and then stimulated or not with IL-1β for 3 h; this is the best time to detect NF-κB subunit activation and translocation. After that, cells were fixed in 4% paraformaldehyde (ThermoFisher Scientific, Milan, Italy) for 15 min and permeabilized in a blocking solution consisting of phosphate-buffered saline (PBS), 1% bovine serum albumin (BSA) (Sigma-Aldrich, Milan, Italy), and 0.2% Triton X-100 (ThermoFisher Scientific, Milan, Italy) for 30 min. Then, chondrocytes were incubated overnight, at 4 °C, with mouse monoclonal anti-NF-κB p50 and anti-NF-κB p65 primary antibodies (dilution 1:100) (Santa Cruz Biotechnology, Milan, Italy), followed by 1 h of incubation with a goat anti-mouse IgG-Texas Red conjugated antibody (dilution 1:100)(Southern Biotechnology, Rome, Italy). After labeling, the slides were washed with PBS and treated with DAPI (Sigma-Aldrich, Milan, Italy) for 10 min. Finally, the slides were mounted with 1,4-diazabicyclo[2.2.2]octane (DABCO) (Sigma-Aldrich, Milan, Italy). OA chondrocytes were observed with Leica DMI 6000 Fluorescence Microscope (Leica Microsystems, Wetzlar, Germany), and the images were obtained using Leica AF6500 Integrated System for Imaging and Analysis (Leica Microsystems, Wetzlar, Germany). A maximum of 100 cells for each experimental condition was considered. Immunocytochemistry staining intensity was scored as absent or from limited to intense; in each group, the scores were represented as the percentages of the total number of the cells [39].

### 2.7. Statistical Analysis

Three different experiments were performed, and the results were reported as mean ± standard deviation (SD) of triplicate values for each experiment. The final normal data distribution was checked and confirmed using Shapiro–Wilk test. All the results were analyzed via a mixed model ANOVA with Bonferroni post hoc test. Linear correlations were analyzed using Spearman’s coefficient. The analyses were carried out using GraphPad Prism version 6.01/b for Windows (GraphPad Software version 6.01/b, Boston, MA, USA, www.graphpad.com, accessed on 10 February 2021). A *p*-value < 0.05 was considered as statistically significant.

## 3. Results

### 3.1. XIST Expression and Its Regulation in IL-1β-Stimulated Chondrocytes

The analysis performed using real-time PCR demonstrated a significant up-regulation of XIST gene expression in OA chondrocytes in comparison to the normal T/C-28a2 cells used as controls (*p* < 0.01, Figure 1a). Moreover, an increase in the apoptosis ratio and up-regulation of MMP-13 mRNA levels was found in OA cells, with a concomitant reduction in the percentage of cell viability and in B-cell lymphoma (BCL2) gene levels (*p* < 0.05, *p* < 0.001, Appendix A).

To investigate the effects of XIST on cell viability, apoptosis, ECM degradation, and miRNA expression, OA chondrocytes were transiently transfected with a specific XIST inhibitor and then exposed or not to the stimulus of IL-1β for a period of 24 h.

Our data showed, first of all, that the expression of XIST was significantly increased (*p* < 0.05) by the stimulus with IL-1β in comparison to the NC, while XIST silencing induced a significant reduction of its gene levels with respect to NC and IL-1β exposure (*p* < 0.05, *p* < 0.01, Figure 1b). Concerning the other analyzed factors, we observed that IL-1β significantly decreased the percentage of survival cells, induced apoptosis and MMP-13 expression, and limited the transcriptional levels of the anti-apoptotic marker BCL2 and miR-146a in comparison to NC (*p* < 0.05, *p* < 0.01, Figure 1c–g). On the other hand, XIST inhibition significantly reduced apoptosis and MMP-13 mRNA, enhanced viability, and up-regulated BCL2 and miR-146a gene levels (*p* < 0.05, Figure 1c–g). Interestingly, after XIST silencing, the negative effects induced by IL-1β were significantly attenuated (*p* < 0.05, *p* < 0.01, Figure 1c–g).

### 3.2. MiR-146a Effects in IL-1β-Stimulated Chondrocytes

To confirm the interplay between miR-146a and XIST function, OA cells were transiently transfected with a specific miR-146a inhibitor before the stimulus with IL-1β for a period of 24 h.

The gene levels of miR-146a were significantly reduced in OA chondrocytes with respect to normal ones (*p* < 0.001, Figure 2a) and after the silencing of OA cells with the miR-146a inhibitor compared to NC and IL-1β exposure (*p* < 0.05, *p* < 0.01, Figure 2b).

Furthermore, IL-1β significantly decreased the percentage of viability and increased the apoptosis and gene expression of MMP-13 and XIST but limited BCL2 mRNA (*p* < 0.05, *p* < 0.01, Figure 2c–g). Similarly, when chondrocytes were silenced with the miR-146a inhibitor, a significant reduction in viability and BCL2 mRNA and an increase in apoptosis and XIST and MMP-13 expression were found in comparison to the NC and IL-1β stimulus (*p* < 0.05, *p* < 0.01, *p* < 0.001, Figure 2c–g). In addition, the concomitant exposure of the cells to miR-146a silencing and IL-1β stimulus significantly exacerbated the negative effects induced by IL-1β with respect to the single treatment (*p* < 0.05, Figure 2c–g).

### 3.3. XIST/miR-146a Axis Modulates NF-κB Signaling

We hypothesized NF-κB as the possible signaling pathway by which XIST and miR-146a interact to exert their opposite effects in the studied mechanisms; for this reason, we performed an evaluation of p50 and p65 NF-κB subunits.

The results of the real-time PCR reported, first of all, increased mRNA levels of p50 and p65 subunits in OA chondrocytes in comparison to T/C-28a2 control cells (*p* < 0.05, Appendix A). Then, OA chondrocytes were transfected with si-XIST and miR-146a inhibitor in the presence of IL-1β (Figure 3a,b), and the analysis demonstrated the up-regulation of both p50 and p65 subunits (*p* < 0.05) in the cells stimulated with IL-1β in comparison to NC. A similar trend was observed after the inhibition of miR-146a in IL-1β-stimulated chondrocytes, with significantly increased gene levels of these factors (*p* < 0.05) when compared to the only negative stimulus with the cytokine. Conversely, the silencing of the cells with si-XIST caused a significant reduction in the transcriptional levels of p50 and p65 (*p* < 0.01), which were up-regulated by IL-1β.

Very interestingly, the modulation of p50 and p65 expression induced by XIST silencing or miR-146a inhibition was significantly attenuated (*p* < 0.01) when the silencing factors were used simultaneously, thus demonstrating the presence of a XIST/miR-146a axis in modulating NF-κB signaling (Figure 3a,b).

To confirm the data obtained in real-time PCR, an immunofluorescence assay of p50 and p65 subunits was carried out, and the results are reported in Figure 4. As expected, the images showed that, in the control condition, the intensity of the signal of p50 and p65 subunits was absent (60%) or limited (38%); only 2% of the chondrocytes had an intense signal (Figure 4a,f), while the label was intense (65%) when the cells were stimulated with IL-1β (Figure 4b,g). The silencing of XIST created a strong reduction in the signal of both subunits (35% limited, 10% intense) activated by IL-1β compared to those induced by the stimulus with the cytokine alone (Figure 4c,h). The inhibition of miR-146a showed a trend similar to that detected in the presence of IL-1β (Figure 4d,i). Finally, an apparent reduction in the signal intensity of p50 and p65 subunits (intense label, respectively, in 45% and 55% of chondrocytes) induced by IL-1β or miR-146a inhibition was found when the inhibitors were used in combination with the silencing of XIST (Figure 4e,j).

### 3.4. XIST and miR-146a Correlations

Figure 5 reports the linear regression models computing the association between XIST and miR-146a with their target genes MMP-13 and BCL2 and their relationship with p50 and p65 subunits of NF-κB. The analysis demonstrated that XIST and miR-146a were negatively correlated (R = −0.385, *p* = 0.0027) (Figure 5a). In addition, we observed that XIST had a positive correlation with MMP-13 (R = 0.2337, *p* = 0.026), p50 (R = 0.2020, *p* = 0.0409) and p65 (R = 0.3143, *p* = 0.0082), while it was negatively correlated with BCL2 (R = −0.3268, *p* = 0.0068) (Figure 5b–e). On the other hand, miR-146a was negatively correlated with the MMP-13 (R = −0.2009, *p* = 0.0416) and p50 subunit (R = −0.1836, *p* = 0.0527) while not closely associated with the BCL2 gene (R = 0.1112, *p* = 0.1397) and p65 subunit (R = −0.04197, *p* = 0.3730) (Figure 5f–i).

## 4. Discussion

As a chronic joint disorder, OA is one of the primary contributors of disability and impairment in the adult and elderly populations [3]. In recent years, extensive efforts have been made in understanding the molecular pathological mechanisms underlying the pathogenesis of the disease. Recent findings have identified lncRNAs as one of the leading players of the OA pathology attributable to their regulatory role in chondrocyte proliferation, apoptosis, and ECM turnover by interacting with different miRNAs [12,18]. Recent evidence affirmed that lncRNA XIST is involved in different kinds of disorders included in OA disease [12]. In particular, it has implications in regulating proliferation and apoptosis and ECM degradation in OA chondrocytes through MAPK and NF-κB pathways [25,26,27]. These regulatory activities of XIST seem to occur via sponging different kinds of miRNAs [25,26,27]. MiR-146a is known to be one of the main miRNAs associated with OA damage because of its role as regulator of cartilage turnover and apoptosis signaling via the NF-κB pathway [40].

Thus, in the current in vitro study on human OA chondrocytes, we explored and better clarified the role of the lncRNA XIST in regulating ECM degradation and apoptosis signaling; furthermore, we hypothesized its negative interaction with miR-146a in modulating these processes through the NF-κB signaling pathway.

The regular turnover of the ECM components is an important hallmark for proper articular cartilage structure and maintenance; this process is mediated by chondrocytes, which produce the main structural proteins as proteoglycans and collagens, and, in the meantime, the proteolytic enzymes responsible for their destruction [41]. During OA development and progression, the balance between anabolism and catabolism of the ECM was lost in favor of catabolic events, which are promoted by a complex network of factors, such as pro-inflammatory cytokines and enzymes active to degrade ECM, as MMPs. MMP-13, also named collagenase 3, is a key regulatory enzyme and a primary catabolic factor strongly expressed during OA damage [41,42]; MMP-13 has the capacity to start the degradation of a large number of downstream matrix and collagen components through specific signaling pathways [41].

Also, the modulation of chondrocyte viability is fundamental for preserving the functionality and stability of the cartilage. Indeed, increased chondrocyte cell death through apoptosis has been associated with OA pathogenesis, participating in the onset and progression of the disease [43]. This process can be activated by different kinds of stimuli, including inflammation, mechanical injury, and mitochondrion dysfunction, inducing an imbalance in releasing pro- and anti-apoptotic elements in the cytosol. Among them, BCL-2 family proteins are a class of anti-apoptotic proteins, and their over-expression protects OA cells from programmed cell death [43,44].

The results of this study showed an increase in the apoptosis ratio and a reduction in the percentage of cell viability in OA primary chondrocytes in comparison to normal cells, used as controls, both in the basal condition and after stimulus with IL-1β, while an up-regulation in MMP-13 and a decrease in BCL2 transcriptional levels were found. As expected, these findings are in line with the data derived from the literature [25,27,42].

It has been demonstrated that the lncRNA XIST is highly expressed in OA tissues with respect to normal samples and that its expression promotes chondrocyte apoptosis and ECM degradation [25,26,27]. According to these data, we showed a significant up-regulation in XIST expression in OA primary chondrocytes compared to normal T/C-28a2 cells, used as controls, and in IL-1β-stimulated chondrocytes. We also observed that XIST silencing significantly reduced apoptosis and MMP-13 expression while enhancing viability and up-regulating the anti-apoptotic marker BCL2, altered by IL-1β. Similarly, Wang et al. [26] revealed that XIST knockdown limited ECM degradation via down-regulation in MMP-13 and disintegrin and metalloproteinase with thrombospondin motif (ADAMTs)-5 expression in IL-1β-treated chondrocytes. Moreover, the regulation in cell proliferation and apoptosis was observed in OA cells incubated with a specific XIST inhibitor through the direct targeting of BCL2 and caspase 3 transcriptional levels [27].

These data confirm the key role of XIST in affecting the viability, apoptosis, and cartilage turnover by a direct effect on BCL2 and MMP-13 target genes, highlighting its contribution to the development and progression of OA.

Accumulating evidence has proved that XIST conducts its regulatory functions by interacting with different miRNAs [25,26,27,45]. For instance, Li et al. [25] first showed the role of the XIST/miR-211 axis in regulating the proliferation and apoptosis of IL-1β-stimulated OA chondrocytes through MAPK signaling [25]. Later, other authors revealed the ability of XIST to regulate ECM degradation in OA cells, controlling the expression of MMP-13 and ADAMTs-5 [26,27]; these studies affirmed the ability of XIST to function as a competitor of miR-1277-5p or miR-149-5p [26,46]. Furthermore, recent findings derived from bioinformatics prediction approaches demonstrated that XIST could target miR-146a, and these data were validated using a luciferase reporter assay [47]. MiR-146a has been widely studied in OA, even if its role in the pathogenesis of the disease is still controversial. Indeed, it has been observed that circulating miR-146a levels were increased in patients with OA in comparison to healthy subjects [48]; similarly, the exosomes of synovial fluid from OA patients were found to contain miR-146a, which is highly expressed in the early stage of the disease while decreasing in its advanced phases [49]. For the cartilage and bone specimens derived from OA patients, the expression of miR-146a was significantly up-regulated in OA cartilage with a low grade on the Mankin scale, while its levels were reduced with disease progression with respect to normal cartilage [50,51,52]. In addition, the expression of miR-146a was down-regulated in OA chondrocytes in comparison to healthy cells [40] and in IL-1β-stimulated cells, where the miRNA rescued the IL-1β-induced inhibition in chondrocyte viability and the expression of the TRAF6-mediated NF-KB signaling pathway [53].

Another result of the present study supports these data, revealing a down-regulation in the miR-146a expression profile in OA chondrocytes and after the IL-1β stimulus in comparison to the relative controls T/C-28a2 and untreated cells. Moreover, we demonstrated that the transfection of cells with the miR-146a-specific inhibitor decreased the percentage of the survival cells and the expression of the anti-apoptotic protein BCL2 while activating apoptosis and up-regulating MMP-13, especially exacerbating the effects caused by IL-1β. Previous studies on OA chondrocytes and synoviocytes confirmed our findings, revealing the involvement of miR-146a in the apoptosis process, inflammation, and cartilage degradation during OA damage via its direct targeting of BCL2, MMP-13, and ADAMTs-5 [40,52,54].

Interestingly, we demonstrated for the first time a direct correlation between XIST and miR-146a. On the one side, we observed that miR-146a inhibition restored the expression of XIST and its negative effects on apoptosis and cartilage turnover in IL-1β-stimulated OA chondrocytes; on the other hand, the silencing of XIST induced opposite effects, promoting the expression of miR-146a while limiting apoptosis and ECM degradation. This evidence prompts us to hypothesize that XIST may exert its detrimental activities in apoptosis and cartilage turnover by regulating miR-146a and its target genes MMP-13 and BCL2. However, these statements need to be confirmed by further data with more robust investigations.

Finally, as the last step, we hypothesized that the effects induced by the XIST/miR-146a axis can be related to the modulation of the NF-κB pathway. Indeed, the current literature attests to the NF-κB pathway as the most renowned mechanism associated with the pathogenesis of different rheumatic diseases, in particular OA [6,55]. Under physiological conditions, NF-κB family proteins, in their inactive form, are sequestered to the cytoplasm through their interaction with the inhibitor of nuclear factor kappa B (IKB)-inhibitory proteins. Upon different kinds of stimuli, such as those induced during OA damage, the activation of the catalytic subunits of the IKK kinase complex induces the degradation of IKB proteins and the release and translocation of NF-κB subunits from the cytoplasm to the nucleus; as a result, NF-κB proteins bind DNA and regulate the transcription of downstream target genes, triggering inflammation, cartilage matrix remodeling, and apoptosis [6,55,56]. In fact, transcriptional and protein expressions of the NF-κB p65 and p50 subunits are highly increased in human OA chondrocytes and in IL-1β-treated cells [6,34,55], as also confirmed by our results.

Furthermore, recent evidence reported that some miRNAs and lncRNAs control NF-κB activation and, in turn, the production of the downstream genes modified by the signaling pathway. In detail, a study performed by Zhou et al. [57] on human OA chondrocytes transfected with a miR-27 mimic or inhibitor highlighted the regulatory activity of this miRNA in inflammatory and degrading activities, both of them crucial in OA pathogenesis, by acting on the NF-κB pathway. Moreover, it was found that miR-146a silencing or over-expression regulated IL-1β-induced apoptosis on OA chondrocytes by targeting NF-KB proteins [53]. Very recently, some authors revealed that the inhibition of XIST fine-tuned cell proliferation and the apoptosis rate by regulating the expression levels of p-p65 and p-IκBα NF-KB in human osteosarcoma cells and in rheumatoid arthritis fibroblast-like synoviocytes [45,58]. According to these findings, in the present study, we demonstrated the regulation of NF-κB via XIST and miR-146a; in particular, we observed that XIST knockdown significantly limited the expressional levels of NF-κB proteins p50 and p65, while miR-146a silencing exerted an opposite impact. Surprisingly, we showed, for the first time, that the effect induced by XIST silencing was partially attenuated by miR-146a inhibition, suggesting their opposite regulation in the NF-κB signaling pathway.

## 5. Conclusions

Our data provide a preliminary basis to increase the state of the art about the importance of LnRNAs, particularly XIST, in mediating the homeostasis of the articular cartilage, probably through exerting an effect on miRNAs. Our findings can be useful to ameliorate the insight about the mechanisms involved in the pathogenesis of OA.

We showed that the knockdown of XIST improved cell viability and limited the apoptosis and ECM destruction of IL-1β-stimulated chondrocytes, revealing the regulatory network among the XIST, miR-146a, and NF-κB pathways; these findings lead us to hypothesize that XIST probably carries out its function in OA through the miR-146a/NF-κB axis.

The most molecular research examples demonstrated that XIST carries out its role as a competing endogenous RNA or a molecular sponge to the downstream miRNAs. For instance, it has been found that XIST acted as a sponge of miR-146a-5p to regulate STAT3 in acute lung injury; the latter accelerated the transcription of XIST, forming a positive feedback loop of STAT3/XIST [47]. Moreover, some authors found that miR-211 can negatively regulate XIST expression and CXCR4 protein levels by directly binding to XIST and CXCR4 30-UTR, respectively; thus, XIST served as a competing endogenous RNA for miR-211 to counteract miR-211-mediated CXCR4 repression [25].

Taken together, our findings lead us to speculate on the role of XIST as a molecular sponge of miR-146a to modify miR-146a-mediated effects on MMP-13 and BCL2, thereby regulating ECM degradation and chondrocyte apoptosis via the NF-κB pathway.

However, this is just a preliminary study, and further investigations are essential to support our hypothesis confirming the presence of the XIST/miR-146a axis. First of all, a dual luciferase reporter assay is necessary to validate our data, demonstrating the interaction between XIST and miR-146a and elucidating their role in OA damage.

The confirmation of the results in healthy primary chondrocytes may be advantageous to elucidate the role of XIST and miR-146a on cell homeostasis, ECM regulation, and, especially, their position in the disease.

In addition, the analysis of the expressional levels of the studied target genes is mandatory to corroborate that the modifications occurring at the mRNA level affected protein modification. Also, a deeper and more accurate analysis of the NF-κB signaling, for instance, using specific NF-κB inhibitors, could be useful to better comprehend the potential implication of the pathway in XIST molecular mechanism.

Although the mechanism through which lncRNAs are able to influence miRNAs is not yet defined, the identification of the competing endogenous lncRNA-miRNA networks may represent a leading goal for the identification of therapeutic targets and the development of targeted therapies.

For instance, the XIST/miR-130a/STAT3 axis has been successfully targeted for the treatment of OA; in particular, the axis has been studied to understand if kaempferol showed protective activities on OA chondrocytes, demonstrating that the compound reduced inflammation and ECM degradation via the XIST/miR-130a/STAT3 axis [59].

Probably, ncRNAs can also act as diagnostic and prognostic markers in different disorders in which the dysregulation in ECM and apoptosis are important hallmark, such as OA; therefore, a possible useful research field could be the identification of ncRNAs markers in the peripheral blood or serum samples of patients with OA to improve the diagnosis of the disease.

## Figures and Tables

**Figure 1 biology-14-00221-f001:**
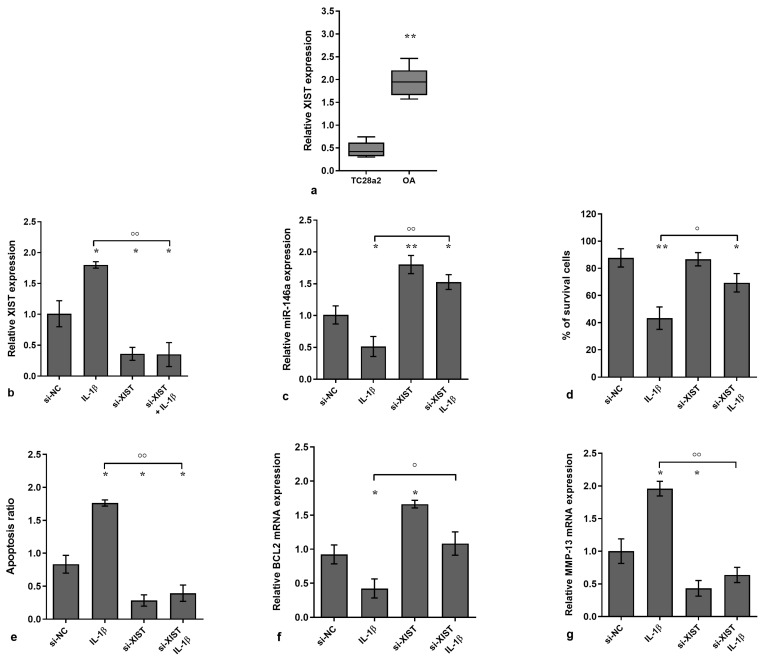
XIST silencing regulates viability, apoptosis, and ECM degradation. Human osteoarthritic (OA) chondrocytes were analyzed in the basal condition after 24 h of transfection with small interfering RNA (siRNA) against XIST (50 nM) or negative controls (NC) (5 nM) in the presence or not of interleukin (IL)-1β (10 ng/mL). (**a**) Gene expression of XIST in T/C-28a2 cell line and in OA chondrocytes was assessed using quantitative real-time PCR. (**b**,**c**,**f**,**g**) Gene expression of XIST, microRNA (miR)-146a, B-cell lymphoma (BCL2), and metalloproteinase (MMP)-13 was assessed using quantitative real-time PCR. (**d**) Cell viability was evaluated using MTT test. (**e**) Apoptosis detection by flow cytometry (Annexin Alexa fluor 488 assay). Data were reported as mean ± standard deviation of triplicate values. * *p* < 0.05, ** *p* < 0.01 versus NC; ° *p* < 0.05, °° *p* < 0.01 versus IL-1β.

**Figure 2 biology-14-00221-f002:**
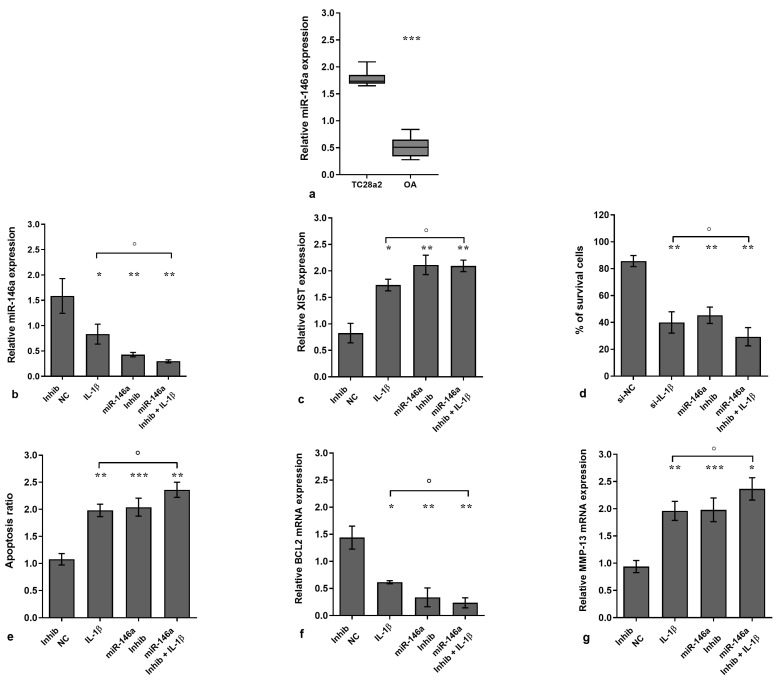
Inhibition of miR-146a reverts the effects of XIST in IL-1β-treated chondrocytes. Human osteoarthritic (OA) chondrocytes were analyzed in the basal condition, after transfection (24 h) with microRNA (miR)-146a-specific inhibitor (50 nM) or negative controls (NC) (5 nM), and in the presence or not of interleukin (IL)-1β (10 ng/mL). (**a**) Gene expression of miR-146a in T/C-28a2 cell line and in OA chondrocytes was assessed using quantitative real-time PCR. (**b**,**c**,**f**,**g**) Gene expression of microRNA (miR)-146a, XIST, B-cell lymphoma (BCL2), and metalloproteinase (MMP)-13 was analyzed using quantitative real-time PCR. (**d**) Cell viability was evaluated using MTT assay. (**e**) Detection of apoptosis via flow cytometry (Annexin Alexa fluor 488 assay). Data were reported as mean ± standard deviation of triplicate values. * *p* < 0.05, ** *p* < 0.01, *** *p* < 0.001 versus NC; ° *p* < 0.05 versus IL-1β.

**Figure 3 biology-14-00221-f003:**
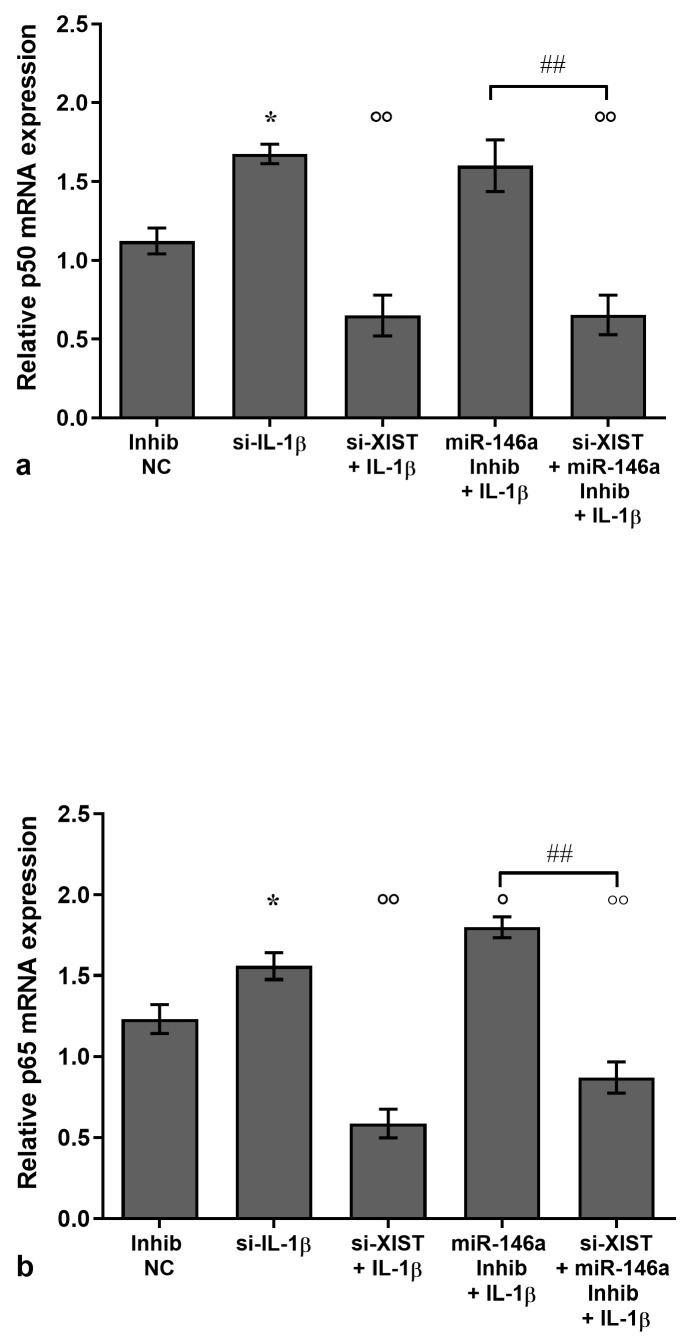
XIST/miR-146a axis on the NF-κB signaling pathway. Human osteoarthritic (OA) chondrocytes were analyzed after 24 h of transfection with small interfering RNA (siRNA) against XIST (50 nM) and/or microRNA (miR)-146a-specific inhibitor (50 nM) or negative controls (NC) (5 nM) in the presence or not of interleukin (IL)-1β (10 ng/mL). (**a**,**b**) Gene expression of p50 and p65 subunits of nuclear factor (NF)-κB was analyzed using quantitative real-time PCR. Data were reported as mean ± standard deviation of triplicate values. * *p* < 0.05 versus NC; ° *p* < 0.05, °° *p* < 0.01 versus IL-1β; ## *p* < 0.01 versus si-XIST.

**Figure 4 biology-14-00221-f004:**
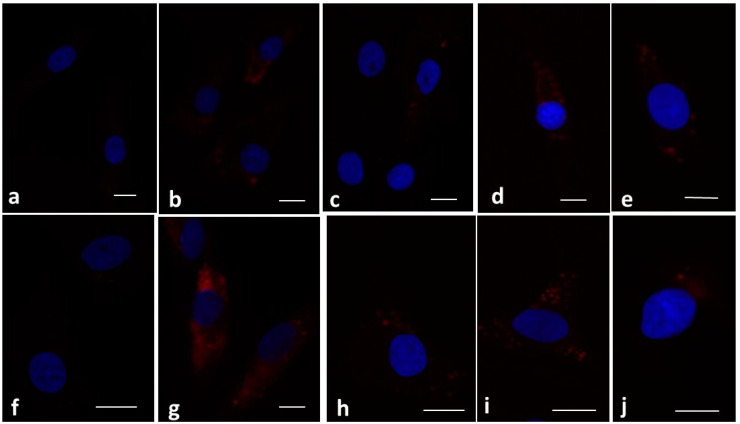
XIST/miR-146a axis of the NF-κB signaling pathway. Human osteoarthritic (OA) chondrocytes were analyzed after 24 h of transfection with small interfering RNA (siRNA) against XIST (50 nM), and/or microRNA (miR)-146a-specific inhibitor (50 nM), or negative controls (NC) (5 nM), and in the presence of interleukin (IL)-1β (10 ng/mL) (3 h). (**a**–**j**) Indirect immunofluorescence microscopy of cells incubated with monoclonal anti-p50 (**a**–**e**) and anti-p65 (**f**–**j**) subunits primary antibodies. Secondary antibody was used: a goat anti-mouse IgG-Texas Red conjugated antibody (red fluorescence). (**a**,**f**) CTRL: limited fluorescence is observed in the cytoplasm; (**b**,**g**) IL-1β: an intense signal is evident in the cytoplasm; (**c**,**h**) si-XIST + IL-1β; the label is limited. (**d**,**i**) miR-146a inhibitor + IL-1β: the signal is diffused in the cytoplasm and not reduced compared to IL-1β. (**e**,**j**) si-XIST + miR-146a inhibitor + IL-1β: the label is strongly reduced. Nuclei (blue) were stained with DAPI. Bars: 50 µm of A-L.

**Figure 5 biology-14-00221-f005:**
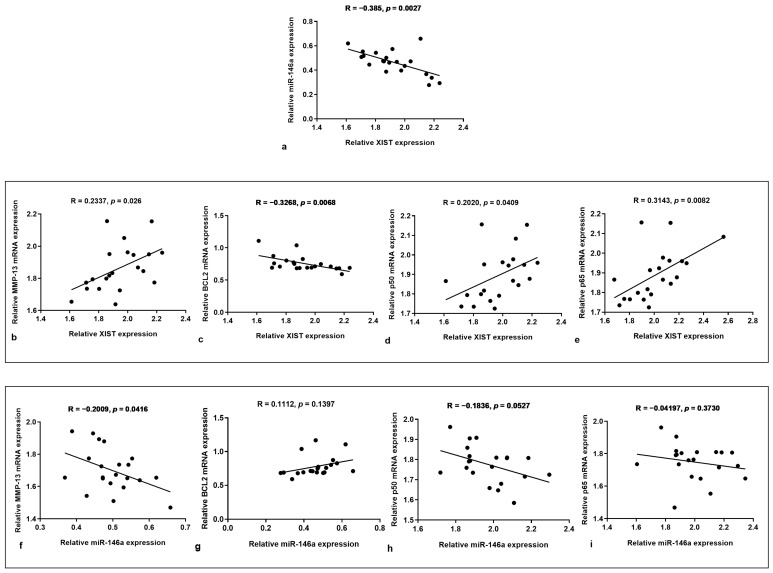
Direct targeting of XIST and miR-146a. (**a**–**i**) The analysis of the linear correlation of XIST, microRNA (miR)-146a, metalloproteinase (MMP)-13, B-cell lymphoma (BCL2), and p50 and p65 subunits of nuclear factor (NF)-κB in human osteoarthritic (OA) chondrocytes was carried out using Spearman’s correlation coefficient.

## Data Availability

The data used and/or analyzed during the current study are available from the corresponding author upon reasonable request.

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
