# Peer review of "Role of lncRNA XIST/miR-146a Axis in Matrix Degradation and Apoptosis of Osteoarthritic Chondrocytes Through Regulation of MMP-13 and BCL2"

_biology, 2025, doi:10.3390/biology14030221_

Round 1
Reviewer 1 Report
Comments and Suggestions for Authors
This study investigates the role of lncRNA XIST in osteoarthritis (OA) chondrocytes. XIST was highly expressed in OA cells and silencing it enhanced cell survival, reduced apoptosis, increased Bcl2 expression, and decreased MMP-13 and NF-κB activation under IL-1β stimulation. The study also revealed that XIST regulates miR-146a, which modulates these effects. These findings highlight the XIST/miR-146a axis as a key player in OA pathogenesis, influencing cell viability, apoptosis, and matrix degradation.
· In introduction and discussion part, clearly articulate the knowledge gap regarding XIST's role in OA progression and then making connection apoptosis, and ECM degradation should be a representation.
· In discussion part, adding explanation why studying the interaction between XIST, miR-146a, and the NF-κB signaling pathway is significant for understanding OA pathology and creating a logical flow by linking the importance of XIST in other diseases to its potential role in OA, should highlight the research focus.
· This manuscript presents a comprehensive study on the role of long non-coding RNA XIST in the regulation of ECM degradation and apoptosis in osteoarthritis (OA) chondrocytes, with a particular focus on its interaction with miR-146a and the NF-κB signaling pathway. The experimental data are robust and align with existing literature, providing valuable insights into the molecular mechanisms underlying OA pathogenesis.
· While the authors suggest that miR-146a plays a significant role in regulating OA progression, its contradictory expression profile in OA tissues (upregulated in circulation but downregulated in cartilage) warrants further discussion. A clearer explanation of the conflicting data could provide a more nuanced understanding of its role in OA.
· The authors propose an interaction between XIST and miR-146a in regulating apoptosis and ECM turnover. Further mechanistic insights into how this interaction occurs at a molecular level, such as through direct binding or competing endogenous RNA mechanisms, would provide deeper understanding.
· The manuscript would benefit from a more detailed discussion of future directions. Specifically, how might these findings translate to therapeutic strategies for OA? It would be helpful to suggest potential clinical applications, such as targeting the XIST/miR-146a axis in OA treatment.
· Overall, the manuscript is well-written and provides significant contributions to understanding the molecular mechanisms involved in OA.
Author Response
We would like to thank the reviewer for the kind appreciation of the manuscript and also for her/his helpful comments, criticisms and suggestions which could be really improved the manuscript. We revised and modified the text, or we have answered to the raised questions, according to her/his requests

Reviewer 2 Report
Comments and Suggestions for Authors
In general, the manuscript is clear and well-written.
My comments are as follows:
A better introduction of the involvement of cartilage/chondrocytes in OA should be added. Considering the journal, it would be important to briefly specify that OA cartilage undergoes to degeneration and both biomechanical and biochemical changes (DOI:10.1016/bs.acc.2020.06.001; DOI: 10.3390/pr11041014 etc).
Lines 22-23: “ The analysis of p50 and p65 nuclear factor (NF)-κB was assessed by PCR an immunofluorescence.” Is unclear.
How did authors select to study the possible interaction between XIST and miRN-146a (among all existing miRNA)?
Lines 80-83: BMI of patients should be added.
Lines 99-103: please add a reference.
Lines 108-112: could the authors add the sequences?
Line 118: here authors reported the use of FCS, while FBS was used in the other experiments. Why?
Lines 181-182: this part should be deleted as it is reported in details at lines 183-186.
Lines 195-197: authors reported that normality was checked using 3 statistical tests (Shapiro–Wilk, D’Agostino and Pearson, and Kol-196 mogorov–Smirnov tests). This is unclear. Did authors perform the three test for each experiment? This point needs clarification.
Figure 1: “si-IL-1beta” is confusing. It seems that authors silenced the cells for ILbeta. Please modify in order to avoid confusion.
Figure 4 is unclear. Authors need to specify the colours. I mean I guess that anti-p50 and p65 are marked in red but it needs to be clearly reported. Please improve the quality of the figures in order to better see red signal.
The stimulation of chondrocytes with Il-1 beta should induce translocation of p65 and p50 in nucleus, but here they are located in cytoplasm (figure 4 b and g). Is that correct? Could the authors explain this point?
It is not clear to me whether the authors believe that XIST interacts with miR-146a. If so, a dual-luciferase reporter assay should be performed. Correlations are not sufficient to demonstrate this interaction.
Other comments:
Please check the whole manuscript for “ml” and substitute with “mL”.
Lines 457: “ should be deleted.
Lines 463: it is unclear to me why authors reported “not applicable” . Authors should report what is stated at lines 85-87:
Author Response

(The authors gave the same response as above.)

Round 2
Reviewer 2 Report
Comments and Suggestions for Authors
No other comments
Author Response
We really thank the reviewer for the kind appreciation of the manuscript and again for her/his helpful comments, criticisms and suggestions which have been really improved the manuscript.
We send to you our very best regards.